# FLATPOSE: AN UPSAMPLING-FREE TRANSFORMER FOR HUMAN POSE ESTIMATION

## ABSTRACT

Human Pose Estimation (HPE) methods often face a trade-off between accuracy and computational cost, which is largely driven by the reliance on upsampling layers to generate high-resolution feature maps. This paper examines the necessity of this convention, investigating whether spatial upsampling is indispensable for precise pose estimation. Our preliminary experiments reveal that coordinate classification-based methods exhibit notable robustness to feature map resolution, unlike their heatmap-based counterparts. This insight suggests a promising, yet challenging, path toward developing entirely upsampling-free architectures. To address the core challenge of recovering fine-grained geometric relationships from spatially coarse features, we introduce FlatPose, a novel and efficient framework that operates directly on low-resolution feature maps from the network backbone. At the heart of FlatPose is a two-stage hierarchical feature enhancement strategy. First, in the Global Encoding stage, we propose the Implicit Coordinate Attention mechanism, which empowers the model to learn a dynamic, content-aware "semantic coordinate system" to model complex, non-local geometric structures from spatially coarse features. Second, in the Targeted Refinement stage, a Salience-Guided selection mechanism identifies the most critical feature regions, which are then deeply optimized via a targeted cross-attention module that focuses computation where it is most needed. Extensive experiments on the challenging COCO, MPII, and CrowdPose benchmarks show that FlatPose achieves a compelling balance between accuracy and computational efficiency. Our work validates that high-precision pose estimation is achievable without explicit upsampling, offering a new and effective paradigm for the field. Our code will be open source.

## 1 INTRODUCTION

Human Pose Estimation (HPE) Chen et al. (2023a); Sun et al. (2019); Zhou et al. (2023) is a fundamental task in computer vision, has largely evolved along two main paradigms, as conceptually illustrated in Figure 1(a). The dominant heatmap-based methods (Paradigm 1) attain high precision by processing high-resolution feature maps, which necessitates computationally expensive upsampling layers (e.g., deconvolution) to restore feature resolution Cai et al. (2020); Wang et al. (2023); An et al. (2024). This reliance introduces a significant bottleneck, increasing both complexity and parameters, which in turn hinders their use in many practical scenarios Janampa & Pattichis (2025); Jiang et al. (2023). This high computational cost prompted us to investigate a critical question: is this reliance on high-resolution features an immutable requirement for all HPE paradigms?

To answer this, we performed a controlled experiment summarized in Figure 1(b). This high computational cost prompted us to investigate a critical question: is this reliance on high-resolution features an immutable requirement for all HPE paradigms? To answer this, we performed a controlled experiment summarized in Figure 1(b). We started from a baseline based on a ResNet50 backbone, using three deconvolution layers to produce a high-resolution $64 \times 48$ feature map, and then progressively reduced the resolution down to $8 \times 6$ by removing these upsampling stages. The results are striking: while the performance of the heatmap-based approach collapses from 71.8 AP to a mere 17.2 AP as resolution decreases, coordinate-based methods demonstrate remarkable robustness. This aligns with the preliminary findings in SimCC Li et al. (2022) and RTMPose Jiang et al. (2023). This growing body of evidence suggests that the coordinate classification paradigm is inherently robust

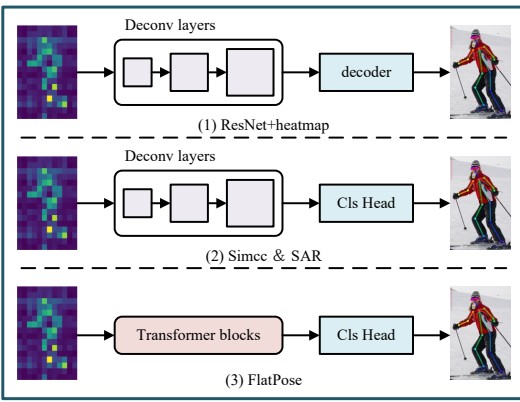 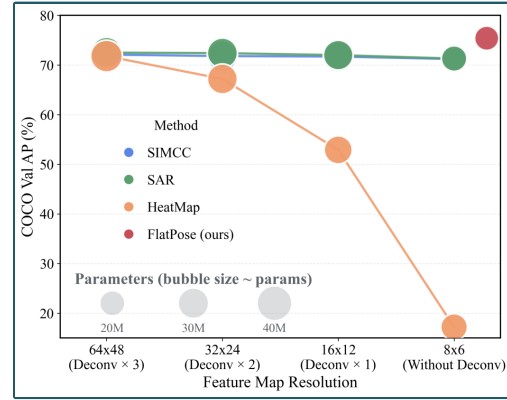

(a) A conceptual comparison of different HPE paradigms

(b) Performance comparison (AP on COCO Val) of different paradigms

Figure 1: Motivation for our upsampling-free approach. (a) A conceptual comparison of different HPE paradigms, showing how FlatPose avoids computationally expensive upsampling layers. (b) Performance comparison on COCO Val under decreasing feature map resolution. The experiment demonstrates the robustness of coordinate-based methods and FlatPose to lower resolutions, in contrast to the collapse of heatmap-based methods, which motivates our design.

to feature map resolution, opening a promising path toward entirely upsampling-free architectures that can alleviate the efficiency burden of conventional designs.

This insight motivates a fundamental architectural shift, away from the upsampling-heavy Paradigm 1 towards the more efficient Paradigm 3 shown in Figure 1(a). We introduce FlatPose, a novel and efficient framework that fully embraces this shift by operating directly on low-resolution feature maps using a unified Transformer architecture. FlatPose employs a hierarchical feature enhancement strategy to achieve precise decoding from coarse features. For the first stage, we propose the Implicit Coordinate Attention (ICA) mechanism, a novel module designed to empower the model with geometric reasoning capabilities directly on coarse feature representations. Unlike methods using static positional encodings Shi et al. (2022); Yuan et al. (2021); Liu et al. (2023), ICA dynamically learns a content-aware "semantic coordinate system," enabling it to model the complex, non-local geometric structures of the human body with high precision. Subsequently, in the refinement stage, a process of Targeted Key Region Refinement identifies the most critical feature regions for each keypoint. These selected features, acting as queries, then interact with the full feature context (serving as keys and values) within a targeted cross-attention module. This interaction allows the model to deeply refine the information within these critical regions. The updated features are then integrated back into the full context, iteratively enhancing the overall representation for a more precise final prediction.

The main contributions of this paper are summarized as follows: 1) Our systematic experiments reveal that coordinate classification methods are significantly more robust to low-resolution features than their heatmap-based counterparts, providing a solid empirical foundation for designing upsampling-free HPE models. 2) We propose FlatPose, an efficient, upsampling-free HPE framework. It features our novel Implicit Coordinate Attention (ICA) module for capturing global geometric relationships and a targeted cross-attention mechanism for refining key features, enabling high-precision localization on low-resolution feature maps. 3) Extensive experiments on challenging benchmarks show that FlatPose is highly competitive, achieving a compelling balance of accuracy and efficiency by significantly reducing computational cost and parameters compared to models that rely on upsampling.

## 2 RELATED WORK

Heatmap-based methods Zhang et al. (2020); Feng et al. (2023); Purkrabek & Matas (2025); Newell et al. (2016) have become a dominant paradigm in human pose estimation due to their superior accuracy. These methods localize keypoints by predicting a 2D spatial probability distribution (a heatmap) for each point. However, their core challenge lies in the computational overhead required

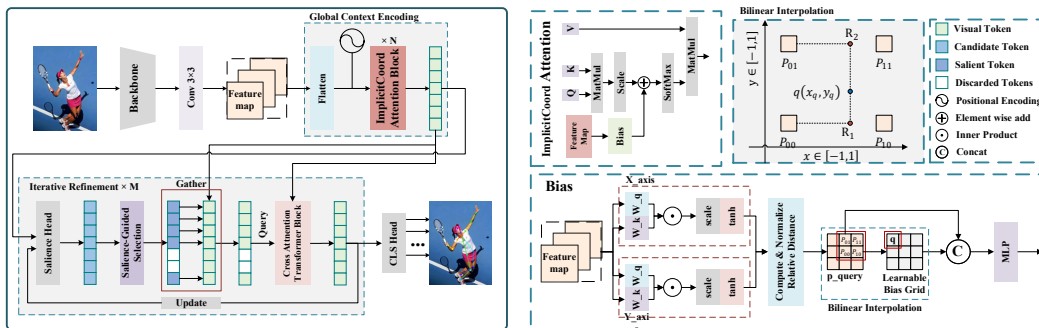

Figure 2: The overall architecture of FlatPose and a detailed illustration of the ImplicitCoord Attention (ICA) mechanism. The left panel shows the main pipeline, where features are first encoded by N blocks and then refined for M iterations. The right panel details the ICA mechanism, where a dynamic geometric bias is generated by interpolating a learnable grid and injected into the attention score.

to generate high-resolution heatmaps. Various strategies have emerged to mitigate this, such as constructing high-resolution features only for relevant regions An et al. (2024), recovering high-quality heatmaps from low-resolution features Wang et al. (2023); Hu et al. (2024), or using more efficient backbones Wang et al. (2022); Xu et al. (2022; 2023). While innovative, these approaches are fundamentally optimizations within the heatmap paradigm.

To overcome the limitations of heatmap-based methods, such as quantization errors and high computational costs, coordinate classification methods like SimCC have gained significant attention. By transforming continuous coordinate prediction into a discrete classification task, these models can effectively process low-resolution features, granting them a natural robustness to resolution changes. This opens a promising avenue for designing efficient, upsampling-free models. Following this trend, methods like RTMPose Lu et al. (2024); Jiang et al. (2023); Yang et al. (2023) and SAR have further explored reducing or eliminating upsampling layers. This body of work validates the potential of the upsampling-free direction but also surfaces a critical challenge: how to capture complex, non-local geometric relationships from low-resolution features where spatial structure is lost. Standard Transformers Dosovitskiy et al. (2020); Liu et al. (2021) often rely on static positional encodings, which are insufficient for this nuanced task. Our work, FlatPose, directly addresses this gap by introducing a novel attention mechanism designed to learn pose-dependent geometric structures from these coarse, spatially-ambiguous features.

## 3 METHODOLOGY

Our proposed method, FlatPose, is founded on a paradigm that directly challenges the conventional reliance on upsampling layers in human pose estimation. As substantiated by our preliminary experiments (Figure 1), the coordinate classification paradigm exhibits remarkable resilience to reductions in feature map resolution, such a property not shared by heatmap-based approaches. Capitalizing on this insight, we designed a streamlined and efficient framework that entirely bypasses explicit spatial upsampling. The overall architecture of FlatPose is depicted in the left part of Figure 2.

### 3.1 AN UPSAMPLING-FREE PARADIGM VIA FEATURE PROCESSING

Unlike traditional models that restore feature resolution, our approach operates directly on the low-resolution feature map produced by the backbone network. Let the initial feature map be $F \in \mathbb{R}^{C \times H_f \times W_f}$, where $C$ is the number of channels, and $H_f, W_f$ are the feature dimensions. First, we use a $3 \times 3$ convolutional layer to project the channel dimension $C$ to match the embedding dimension $D$ required by our Transformer modules.

$$F' = \text{Conv}_{3 \times 3}(F) \tag{1}$$

Here, the resulting feature map $F' \in \mathbb{R}^{D \times H_f \times W_f}$ maintains its spatial dimensions but has an adjusted channel depth $D$. Subsequently, we reshape this 2D feature map $F'$ into a 1D sequence of

tokens $X \in \mathbb{R}^{L \times D}$, where the sequence length $L = H_f \times W_f$. To preserve spatial awareness after this process, we add a standard 2D sinusoidal position encoding $E_{\text{pos}} \in \mathbb{R}^{L \times D}$ to the token sequence, yielding the initial input $X_0$ for our feature enhancement pipeline.

$$X_0 = X + E_{\text{pos}} \tag{2}$$

This initial processing is the foundation of our upsampling-free framework. All subsequent geometric and contextual relationship modeling is performed on this 1D token representation.

## 3.2 Global Context Encoding with Implicit Coordinate Attention (ICA)

Token sequences inherently lack the explicit 2D structure required for many vision tasks. To overcome this, we propose Implicit Coordinate Attention (ICA), a mechanism that embeds geometric reasoning directly into the attention process, as detailed in the right part of Figure 2. ICA distinguishes itself from prior work in two main aspects. In contrast to Deformable Attention Zhu et al. (2020), which learns sparse *sampling offsets*, ICA computes a *dense bias matrix* to augment the standard attention scores. Furthermore, while Conditional Positional Encodings (CPE) Chu et al. (2021) generate content-aware *absolute* position embeddings, ICA's bias is a function of the *relative semantic distance* between tokens. This bias is generated by a continuous function, learned through feature grid interpolation, which uniquely allows the model to reason explicitly about geometric relationships.

### 3.2.1 Dynamic Semantic Coordinate Generation

Instead of relying on fixed positions, ICA dynamically computes a coordinate for each feature token based on its semantic content. As shown in the "Bias" generation process in Figure 2, this begins by projecting the input token sequence $X_0 \in \mathbb{R}^{L \times D}$ into specialized query-like and key-like representations for both the X and Y axes.

$$Q_{r,x} = X_0 W_{q,x} \qquad\qquad Q_{r,y} = X_0 W_{q,y} \tag{3}$$
$$K_{r,x} = X_0 W_{k,x} \qquad\qquad K_{r,y} = X_0 W_{k,y} \tag{4}$$

Here, $W_{q,x}, W_{k,x}, W_{q,y}, W_{k,y} \in \mathbb{R}^{D \times D'}$ are learnable projection matrices, and we set the intermediate dimension to $D' = D/4$. The semantic coordinate $c_x^{(i)}$ for the $i$-th token along the x-axis is then generated via a scaled inner product passed through a tanh activation.

$$c_x^{(i)} = \tanh\left(\gamma \langle q_{r,x}^{(i)}, k_{r,x}^{(i)} \rangle\right) \tag{5}$$

In this equation, the inner product $\langle \cdot, \cdot \rangle$ computes a content-based similarity score. This score is scaled by a learnable parameter $\gamma$ and mapped to the range $[-1, 1]$ to produce the final coordinate. An identical process is performed independently to compute the corresponding semantic coordinate $c_y^{(i)}$ along the y-axis.

### 3.2.2 Continuous Relative Positional Bias Generation

Having established the semantic location of each token, the attention bias is generated as a continuous function of their relative semantic distance. This process involves several steps, as depicted in the right part of Figure 2.

First, for any pair of tokens $(i, j)$, we compute their semantic distance vector $\Delta c^{(i,j)}$.

$$\Delta c^{(i,j)} = (|c_x^{(i)} - c_x^{(j)}|, |c_y^{(i)} - c_y^{(j)}|) \tag{6}$$

Since $c_x$ and $c_y$ are in $[-1, 1]$, the elements of $\Delta c^{(i,j)}$ are in $[0, 2]$. We then normalize this vector to a query coordinate $p_{\text{query}}^{(i,j)}$ in the range $[-1, 1]^2$ for grid sampling. This combined step corresponds to the "Compute & Normalize Relative Distance" block in the diagram.

$$p_{\text{query}}^{(i,j)} = \Delta c^{(i,j)} - 1 \tag{7}$$

This query coordinate is used to probe a small, learnable bias grid $G_{\text{bias}} \in \mathbb{R}^{D_{\text{bias}} \times S \times S}$ via differentiable bilinear interpolation, denoted by $\phi$.

$$f_{\text{interp}}^{(i,j)} = \phi(G_{\text{bias}}, p_{\text{query}}^{(i,j)}) \tag{8}$$

Table 1: Detailed configurations of our FlatPose model variants. The 'Coarse' and 'Refine' columns denote the number of coarse-tuning and fine-tuning blocks, respectively. The backbone is based on ConvNeXtV2 Woo et al. (2023).

| Model | Input | Backbone | Dim. | Coarse | Refine | K | Params(M) | GFLOPs | AP |
|-------|-------|----------|------|--------|--------|---|-----------|--------|-----|
| FlatPose-B | $256 \times 192$ | ConvNeXtV2-N | 256 | 4 | 2 | 4 | 21.6 | 2.8 | 75.4 |
| FlatPose-L | $256 \times 192$ | ConvNeXtV2-T | 256 | 6 | 3 | 4 | 37.3 | 4.9 | 77.4 |
| FlatPose-L | $384 \times 288$ | ConvNeXtV2-T | 256 | 6 | 3 | 6 | 37.4 | 11.5 | 78.2 |



Figure 3: Illustration of the Salience-Guided selection strategy. Salience scores are computed for each feature token. The tokens with the top-K highest scores are selected as queries for the refinement stage.

As detailed in the "Bilinear Interpolation" panel of Figure 2, this step samples a feature vector $f_{\text{interp}}$ for each continuous query coordinate. This feature is then concatenated with the precise query coordinate and refined by an MLP to produce the final bias $B^{(i,j)}$ for each of the $N_h$ attention heads.

$$B^{(i,j)} = \text{MLP}_{\text{interp}}(\text{concat}[f_{\text{interp}}^{(i,j)}, p_{\text{query}}^{(i,j)}]) \tag{9}$$

### 3.2.3 FINAL ATTENTION INTEGRATION

The final attention score is computed by integrating the generated bias with standard content-based attention. As shown in the ICA overview in Figure 2, the bias term $B$ is added to the scaled dot-product scores of the content-based query ($Q_{\text{attn}}$) and key ($K_{\text{attn}}$).

$$\text{Score}(i,j) = \frac{Q_{\text{attn}}^{(i)}(K_{\text{attn}}^{(j)})^T}{\sqrt{d_k}} + B^{(i,j)} \tag{10}$$

Here, $d_k$ is the dimension of the key vectors. This combined score allows the model to consider both content similarity and learned geometric priors.

### 3.3 ITERATIVE REFINEMENT WITH TARGETED CROSS-ATTENTION

After the Global Context Encoding stage establishes a general understanding of the pose, we introduce an Iterative Refinement stage to achieve high precision. As illustrated in Figure 2, this stage executes a select-attend-update loop for $M$ iterations. This process progressively enhances the feature representation by focusing computation on the most salient visual tokens. The context entering each iteration is a complete set of visual tokens, denoted as $X_{\text{context}} \in \mathbb{R}^{L \times C}$.

### 3.3.1 SALIENCE-GUIDED SELECTION

The first step in each iteration is to select a small subset of tokens to act as the query. A Salience Head first processes the full set of input visual tokens, $X_{\text{context}}$, assigning a relevance score to each one. This is conceptually similar to token pruning strategies in efficient Transformers Xia et al. (2022); Liang et al. (2022); Wang et al. (2024); Chen et al. (2023b). Following this, a selection mechanism identifies the indices $\mathcal{I}$ of the tokens with the highest scores, as illustrated in Figure 3. These indices are then used to collect the corresponding feature vectors from the full context map to form the query. This selection process is formally defined as:

$$X_{\text{query}} = \text{Gather}(X_{\text{context}}, \mathcal{I}) \tag{11}$$

Here, the Gather operation selects tokens from the input tensor $X_{\text{context}} \in \mathbb{R}^{L \times C}$ along its first dimension based on the index set $\mathcal{I}$. The resulting query tensor, $X_{\text{query}} \in \mathbb{R}^{L_q \times C}$, is a compact subset of the original context, where $L_q = |\mathcal{I}|$ is the number of selected tokens. To ensure computational efficiency, our design makes $L_q$ significantly smaller than $L$ ($L_q \ll L$).

### 3.3.2 TARGETED REFINEMENT

The compact query sequence $X_{\text{query}}$ is then deeply refined. It is fed as the Query into a Cross Attention Transformer Block, while the full, un-altered context map $X_{\text{context}}$ serves as the source for both the Key and Value. This allows the small set of critical query tokens to attend to the entire global context, efficiently enriching their features. The interaction is computed as:

$$X_{\text{refined}} = \text{CrossAttentionBlock}(X_{\text{query}}, X_{\text{context}}, X_{\text{context}}) \qquad (12)$$

The output, $X_{\text{refined}}$, has the same dimension as the input query, $\mathbb{R}^{L_q \times C}$, but its features are now contextually enriched.

### 3.3.3 CONTEXT UPDATE AND ITERATION

The final step is to integrate the refined features back into the global context. The Update operation is a scatter-overwrite mechanism: the refined tokens in $X_{\text{refined}}$ are written back into the full context map at their original locations, replacing the previous features. This produces an updated context map for the next iteration, $X_{\text{context}}^{(i+1)}$, whose dimensions remain $\mathbb{R}^{L \times C}$. This entire select-attend-update cycle is repeated $M$ times, with each iteration further enhancing the precision of the feature representation.

### 3.3.4 CLS HEAD FOR FINAL PREDICTION

After $M$ refinement iterations, the final enhanced context map is passed to the CLS Head. This head employs two separate linear classifiers to predict the probability distributions for the x and y coordinates of each keypoint.

## 4 EXPERIMENTS

### 4.1 EXPERIMENT SETTINGS

#### 4.1.1 DATASETS AND EVALUATION METRICS.

We conduct extensive experiments on three challenging public benchmarks: MS COCO Lin et al. (2014), MPII Human Pose Andriluka et al. (2014), and CrowdPose Li et al. (2019), using their standard evaluation protocols. For COCO, we train on 'train2017' and evaluate on 'val2017' and 'test-dev2017' using mean Average Precision (AP). For MPII, we report the head-normalized Percentage of Correct Keypoints (PCKh@0.5). For CrowdPose, we use AP to evaluate robustness in crowded scenes.

#### 4.1.2 IMPLEMENTATION DETAILS.

Models are trained for 210 epochs on 8 NVIDIA RTX 4090 GPUs using a ConvNeXt backbone and the AdamW optimizer with a base learning rate of $1 \times 10^{-3}$. We use a cosine annealing schedule with a linear warm-up. Standard data augmentation and flip-testing at inference are employed. An Exponential Moving Average (EMA) hook is used to stabilize training.

#### 4.1.3 MODEL CONFIGURATIONS

To demonstrate scalability, we instantiate several FlatPose variants by varying the backbone and module depths. Detailed configurations are presented in Table 1.

Table 2: Comparison with state-of-the-art methods on the COCO validation set. FlatPose-L is directly compared to SHaRPose-Base, with performance and efficiency changes indicated by arrows (↑↓). Best results for each metric are in bold.

| Method | Input | Params(M) | GFLOPS | AP | AP$^{50}$ | AP$^{75}$ | AR |
|---|---|---|---|---|---|---|---|
| SimpleBaseline Xiao et al. (2018) | $256 \times 192$ | 68.6 | 15.7 | 72.0 | 89.3 | 79.8 | 77.8 |
| HRNet-W48 Sun et al. (2019) | $256 \times 192$ | 63.6 | 14.6 | 75.1 | 90.6 | 82.2 | 80.4 |
| TokenPose-B Li et al. (2021) | $256 \times 192$ | 13.5 | 5.7 | 74.7 | 89.8 | 81.4 | 80.0 |
| TokenPose-L/D24 Li et al. (2021) | $256 \times 192$ | 27.5 | 11.0 | 75.8 | 90.3 | 82.5 | 80.9 |
| ViTPose-S Xu et al. (2022) | $256 \times 192$ | 24.3 | 5.6 | 73.9 | 90.3 | 81.6 | 79.2 |
| ViTPose-B Xu et al. (2022) | $256 \times 192$ | 89.9 | 18.5 | 75.7 | 90.5 | 82.9 | 80.9 |
| SHaRPose-Small An et al. (2024) | $256 \times 192$ | 28.4 | 4.9 | 74.2 | 90.2 | 81.8 | 79.5 |
| SHaRPose-Base An et al. (2024) | $256 \times 192$ | 93.9 | 17.1 | 75.5 | 90.6 | 82.3 | 80.8 |
| RTMPose Jiang et al. (2023) | $256 \times 192$ | 27.7 | 4.2 | 74.8 | - | - | - |
| SimCC Li et al. (2022) | $256 \times 192$ | 66.3 | 14.6 | 75.9 | - | - | **81.2** |
| FlatPose-B (Ours) | $256 \times 192$ | 21.6 | 2.8 | 75.4 | 93.6 | 82.7 | 78.4 |
| **FlatPose-L (Ours)** | $256 \times 192$ | **37.3↓60%** | **4.9↓71%** | **77.4↑1.9** | **93.6** | **84.9** | 80.2 |
| HRNet-W48 Sun et al. (2019) | $384 \times 288$ | 63.6 | 32.9 | 76.3 | 90.8 | 82.9 | 81.2 |
| ViTPose-B Xu et al. (2022) | $384 \times 288$ | 89.9 | 44.1 | 76.9 | 90.9 | 83.2 | 82.1 |
| SHaRPose-Base An et al. (2024) | $384 \times 288$ | 93.9 | 32.9 | 77.4 | 91.0 | 84.1 | **82.4** |
| **FlatPose-L (Ours)** | $384 \times 288$ | **37.4↓60%** | **11.5↓65%** | **78.2↑0.8** | **93.7** | **85.1** | 80.9 |

## 4.2 RESULTS ON COCO VAL SET

Table 2 shows our results on the COCO validation set, where FlatPose demonstrates a superior accuracy-efficiency trade-off. At an input resolution of $256 \times 192$, our lightweight FlatPose-B model achieves a competitive 75.4 AP with only 2.8 GFLOPS. This showcases a remarkable efficiency, as it outperforms recent lightweight models like RTMPose-m by 0.6 AP while using 33% fewer GFLOPS. Our premier model, FlatPose-L, sets a new state of the art for this resolution with 77.4 AP. This result surpasses heavyweight models like ViTPose-B by 1.7 AP and SHaRPose-Base by 1.9 AP, and also outperforms strong coordinate-based methods such as SimCC by 1.5 AP. Crucially, this superior accuracy is achieved with a fraction of the computational cost; its 4.9 GFLOPS represent a 73% reduction compared to ViTPose-B and a 66% reduction compared to SimCC. At the higher $384 \times 288$ resolution, FlatPose-L again achieves a top performance of 78.2 AP, beating previous methods with a 65% reduction in computational cost. As shown in Figure 4, the model's learned attention maps confirm its ability to adaptively focus on relevant body regions, which contributes to the precise final pose.

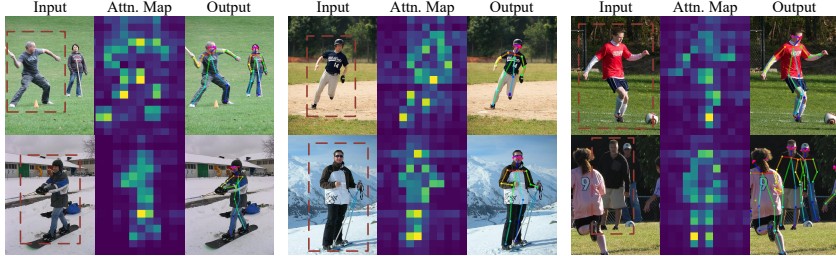

Figure 4: This figure visualizes our model's learned attention maps, demonstrating its ability to adaptively focus on relevant body regions for precise final pose estimation.

## 4.3 RESULTS ON COCO TEST-DEV SET

On the challenging COCO test-dev set, our FlatPose-L model confirms its strong generalization and efficiency, as shown in Table 3. It achieves a state-of-the-art 76.7 AP, outperforming a series of strong contenders. Notably, it surpasses the recently proposed SAR by 0.4 AP, as well as established methods like ViTPose-B by 0.5 AP and SIMCC by 0.7 AP. Crucially, this top-tier accuracy

Table 3: Comparison with state-of-the-art methods on the COCO test-dev set. All models use an input size of $384 \times 288$. Our FlatPose-L is directly compared to ViTPose-B. Best results for each metric are in bold.

| Method | Input | Params(M) | GFLOPS | AP | AP$^{50}$ | AP$^{75}$ | AR |
|---|---|---|---|---|---|---|---|
| SimpleBaseline Xiao et al. (2018) | $384 \times 288$ | 68.6 | 35.6 | 73.7 | 91.9 | 81.1 | 79.0 |
| HRNet-W48 Sun et al. (2019) | $384 \times 288$ | 63.6 | 32.9 | 75.5 | 92.5 | 83.3 | 80.5 |
| SIMCC Li et al. (2022) | $384 \times 288$ | 66.3 | 32.9 | 76.0 | 92.4 | 83.5 | 81.1 |
| SAR Wang & Zhang (2024) | $384 \times 288$ | - | - | 76.3 | 92.5 | 83.6 | 81.2 |
| TokenPose-L/D24 Li et al. (2021) | $384 \times 288$ | 29.8 | 22.1 | 75.9 | 92.3 | 83.4 | 80.8 |
| ViTPose-B Xu et al. (2022) | $384 \times 288$ | 89.9 | 44.1 | 76.2 | 92.7 | 83.7 | 81.3 |
| **FlatPose-L (Ours)** | $384 \times 288$ | **37.4↓58%** | **11.5↓74%** | **76.7↑0.5** | **92.8** | **84.3** | **81.7** |

is delivered with only 11.5 GFLOPs and 37.4M parameters, representing massive 74% and 58% reductions in computation and model size, respectively, when compared to ViTPose-B.

## 4.4 Results on CrowdPose and MPII

On the challenging CrowdPose and MPII benchmarks, FlatPose demonstrates strong robustness and generalization. The comprehensive results are presented in Table 4. On the CrowdPose test set, our FlatPose-L model achieves a competitive performance of 67.9 AP, outperforming methods like ViTPose-B and SARPose, and showing an improvement over the strong HRNet-W32 baseline. For the MPII validation set, FlatPose-L achieves a leading result of 90.7 PCKh@0.5, surpassing other strong methods such as GatedUniPose and HRNet. It is worth noting that all results reported for the MPII dataset were obtained using a $256 \times 256$ input size.

Table 4: Comparison with state-of-the-art methods on the CrowdPose and MPII datasets. For methods not evaluated on a specific dataset, results are marked with '-'.

| Method | CrowdPose | | | | MPII |
|---|---|---|---|---|---|
| | AP | AP$^{(E)}$ | AP$^{(M)}$ | AP$^{(H)}$ | PCKh@0.5 |
| HRNet-W32 Sun et al. (2019) | 67.5 | 77.0 | 68.7 | 55.3 | 90.0 |
| ViTPose-B Xu et al. (2022) | 66.5 | 76.1 | 67.9 | 54.6 | - |
| SRPose Wang et al. (2023) | 64.7 | 74.4 | 65.7 | 52.3 | - |
| SARPose Wang & Zhang (2024) | 66.3 | 73.7 | 63.0 | **57.6** | - |
| SimpleBaseline Xiao et al. (2018) | - | - | - | - | 88.2 |
| SimCC Li et al. (2022) | - | - | - | - | 90.0 |
| TokenPose Li et al. (2021) | - | - | - | - | 89.4 |
| GatedUniPose Feng et al. (2024) | - | - | - | - | 90.2 |
| **FlatPose-L (Ours)** | **67.9** | **78.6** | **69.4** | 54.0 | **90.7** |

## 4.5 Ablation Study

To analyze the contribution of each key component and validate our design choices, we conduct a comprehensive series of ablation studies on the COCO validation set. All experiments are based on our FlatPose-B configuration unless otherwise specified. The results are consolidated in Table 5.

**Analysis of Core Design Choices.** Part A of Table 5 validates our fundamental design decisions. First, to isolate the contribution of our head architecture from the backbone, we replaced Con­vNeXtV2 with a standard ResNet-50. As shown in row (2), our FlatPose head with a ResNet-50 backbone achieves 73.9 AP, significantly outperforming a standard upsampling-based SimpleBase­line (row 1) which scored 72.0 AP. This result confirms the effectiveness of our proposed head, which improves performance by 1.9 AP while simultaneously using 16% fewer GFLOPs. Second, we tested an alternative prediction paradigm by replacing our SimCC classification head with a di­rect regression head trained with L1 loss (row 3). This led to a substantial 1.1 AP drop from 75.4 to

Table 5: Comprehensive ablation studies of FlatPose on the COCO validation set. We analyze the core design choices, attention mechanism, architecture depth, and refinement strategy. The baseline for most ablations is our FlatPose-B model.

| Group | Configuration | Params (M) | GFLOPS | AP |
|-------|---------------|-----------|--------|-----|
| *Part A: Core Design Choices* | | | | |
| (1) | SimpleBaseline (ResNet-50 + Deconv Head) | 34.0 | 5.5 | 72.0 |
| (2) | FlatPose (ResNet-50 Backbone) | 33.4 | 4.6 | 73.9 |
| (3) | FlatPose-B w/ Regression Head (L1 Loss) | 21.9 | 2.9 | 74.3 |
| (4) | FlatPose-B + Upsampling Layers | 24.8 | 3.9 | 75.5 |
| **(5)** | **FlatPose-B (ConvNeXtV2-N, Ours)** | **21.6** | **2.8** | **75.4** |
| *Part B: Attention Mechanism* | | | | |
| (6) | FlatPose-B w/o ICA (Standard Attention) | 21.3 | 2.7 | 74.1 |
| **(5)** | **FlatPose-B w/ ICA (Ours)** | **21.6** | **2.8** | **75.4** |
| *Part C: Architecture Depth (Coarse + Refine)* | | | | |
| (7) | 3 + 3 Blocks | 21.5 | 2.8 | 75.2 |
| (8) | 4 + 4 Blocks | 23.2 | 2.9 | 75.5 |
| **(9)** | **4 + 2 Blocks (Ours)** | **21.6** | **2.8** | **75.4** |
| *Part D: Refinement Strategy (K value)* | | | | |
| (10) | K = 2 | 21.6 | 2.8 | 75.2 |
| (11) | K = 8 | 21.6 | 2.8 | 75.3 |
| (12) | K = Full | 21.6 | 2.8 | 75.4 |
| **(13)** | **K = 4 (Ours)** | **21.6** | **2.8** | **75.4** |

74.3, demonstrating that the coordinate classification scheme is crucial for maintaining high performance in our upsampling-free framework. Finally, to justify our upsampling-free design, we created a variant by adding upsampling layers after our Transformer blocks (row 4). While this slightly increased performance by 0.1 AP (from 75.4 to 75.5), it came at the cost of a nearly 40% surge in GFLOPs, confirming that our upsampling-free approach provides a much better accuracy-efficiency trade-off.

**Analysis of Model Components.** Part B, C, and D analyze the specific components of our model. In Part B, replacing our proposed Implicit Coordinate Attention (ICA) with standard self-attention (row 6) causes a 1.3 AP drop with negligible changes in complexity, highlighting the importance of ICA's learned geometric priors. Part C explores the optimal depth and configuration of our coarse and refine stages. While the 4+4 configuration (row 8) achieves the highest AP at 75.5, the 4+2 block configuration (row 9) achieves a very close 75.4 AP with fewer parameters. We therefore adopt the 4+2 structure as it provides the best balance of performance and complexity. Lastly, Part D studies the impact of the number of selected tokens ($K$) in the refinement stage. Choosing $K = 4$ (row 13) achieves the same peak performance as attending to all tokens ("Full", row 12), but with significantly lower computational overhead in the refinement stage, making it the optimal choice.

## 5 CONCLUSION

In this paper, we addressed the high computational cost associated with the upsampling layers common in human pose estimation. We proposed FlatPose, a novel and entirely upsampling-free framework that operates directly on low-resolution feature maps. Our approach is built upon the robustness of the coordinate classification paradigm and introduces a two-stage feature enhancement strategy. The core of our method is the novel Implicit Coordinate Attention (ICA), which learns a dynamic "semantic coordinate system" to effectively capture complex geometric relationships from flattened features. This is complemented by an iterative refinement stage using salience-guided cross-attention, which efficiently focuses computation on salient regions to achieve high precision. Extensive experiments on challenging benchmarks, including COCO, MPII, and CrowdPose, demonstrate that FlatPose achieves a highly competitive trade-off between accuracy and efficiency, outperforming strong baselines while using significantly fewer computational resources. Ultimately, this work demonstrates that high-accuracy pose estimation does not solely depend on spatial upsampling, providing an effective and promising new paradigm for efficient HPE model design.

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

# LLM USAGE DISCLOSURE

Large language models (LLMs) were used in this work to assist with translation and language refinement. The initial manuscript was drafted by the authors in Chinese and subsequently translated into English with the assistance of an LLM. Furthermore, LLMs were utilized for language polishing to improve grammar, clarity, and sentence structure. The authors carefully reviewed, edited, and verified all LLM-generated text to ensure it accurately reflects their original scientific intent. The authors are solely responsible for the accuracy of all statements, the correctness of the code, and the validity of the results.

**Human verification & responsibility.** All scientific claims, equations, and conclusions were originally conceived and formulated by the authors. All experiments were conducted by the authors in clean environments, and all plots and tables were generated from their verified outputs. The authors manually checked all references to ensure they correspond to real and relevant sources.

**Confidentiality & ethics.** No confidential third-party material (e.g., peer reviews, private manuscripts) was provided to any LLM. We did not include hidden prompt-injection text in the submission. All external data and code used in the experiments obey their respective licenses.

This disclosure is mirrored in the submission form as required by ICLR policy.

## A   APPENDIX: ANALYSIS ON THE ROBUSTNESS OF COORDINATE CLASSIFICATION TO FEATURE RESOLUTION

The disparate reliance on feature upsampling between heatmap-based and coordinate classification-based paradigms stems from fundamental differences in their strategies for encoding and decoding keypoint location information. This difference becomes visually evident when comparing the quality of the final (or equivalent) heatmaps as the spatial resolution of the input feature maps changes, as illustrated in Figure 5.

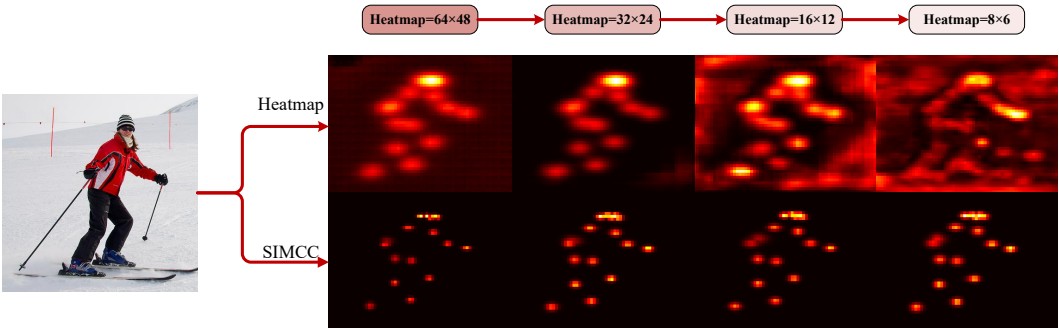

Figure 5: A visualization comparing the generated (or equivalently reconstructed) heatmaps from the two methods as the input feature map resolution is progressively reduced. Notably, even as the resolution of the feature map fed into the SimCC head decreases significantly, the reconstructed equivalent heatmap maintains clear and localizable peaks. While the sharpness or confidence of these peaks may slightly diminish, they do not suffer the catastrophic degradation seen in the traditional heatmap approach.

**Heatmap-based Methods** explicitly encode the location of a keypoint $k$ into a 2D spatial probability distribution, the heatmap $M_k \in \mathbb{R}^{H_{out} \times W_{out}}$. The localization precision is directly coupled with the heatmap's spatial resolution $H_{out}, W_{out}$. When the heatmap is generated from a low-resolution feature map $F \in \mathbb{R}^{C \times H_f \times W_f}$ from the backbone (i.e., $H_{out} \approx H_f, W_{out} \approx W_f$), the effective stride $S = H_{image}/H_f$ is large. This means a single pixel in the heatmap corresponds to a large $S \times S$ region in the original image, introducing significant **quantization error**. This coarse discretization limits the heatmap's ability to represent sub-pixel locations accurately, losing high-frequency spatial details. From an information theory perspective, this constrains the "spatial bandwidth" available for encoding the precise location. Therefore, upsampling is critical for

heatmap-based methods, as it generates higher-resolution heatmaps ($H'_{out} \gg H_f, W'_{out} \gg W_f$), thereby increasing the capacity for spatial encoding to carry finer location information and maximizing the mutual information $I(M_k; L_{true})$ between the predicted heatmap and the true keypoint location.

**Coordinate Classification Methods (e.g., SimCC)** , in contrast, adopt a different encoding strategy. They decouple the 2D localization problem into two independent 1D classification tasks, predicting the coordinates for the X and Y axes separately. The outputs are two probability vectors, $P_X \in \mathbb{R}^{W_{bins}}$ and $P_Y \in \mathbb{R}^{H_{bins}}$, where $W_{bins}$ and $H_{bins}$ are the number of predefined discrete coordinate "bins". The core advantage of this approach is that the resolution of the output coordinates—determined by $W_{bins}$ and $H_{bins}$, which can be set to be much larger than $W_f$ and $H_f$—is **effectively decoupled** from the spatial resolution of the input feature map $F$. The model learns a mapping function $g : F \mapsto (P_X, P_Y)$. Even if the input feature $F$ is spatially coarse (i.e., small $W_f, H_f$ without upsampling), as long as it retains sufficient discriminative cues (often encoded in the channel dimension $C$), a sufficiently powerful mapping function $g$ (which in FlatPose is enhanced by Transformer blocks) can effectively transform these cues into fine-grained 1D probability distributions. The final coordinate is typically decoded by calculating the expected value, e.g., $x_k = \sum_{i=0}^{W_{bins}-1} i \cdot P_X(i)$, which inherently provides a form of sub-pixel interpolation. Therefore, the information bottleneck in the coordinate classification paradigm lies more in the effectiveness of extracting discriminative information from $F$ (i.e., $I(F; L_{true})$) and the capacity of the mapping function $g$, rather than the spatial "bandwidth" limitation of the output representation itself.

In summary, the strong dependency of heatmap-based methods on upsampling arises from their spatial encoding of positional information. In contrast, coordinate classification methods, through information transformation and the decoupling of output and input spaces, demonstrate the potential for high-precision localization without the need for explicit spatial upsampling, providing the key theoretical underpinning for efficient models like FlatPose.

## B APPENDIX: QUALITATIVE VISUALIZATION RESULTS ON COCO

To provide an intuitive understanding of our model's performance, Figure 6 presents qualitative results on challenging examples from the COCO test-dev set. These visualizations showcase FlatPose's ability to accurately localize keypoints even in complex scenarios involving varied poses, scales, and occlusions. The results align with our quantitative findings, demonstrating the robustness and high precision of our upsampling-free approach.

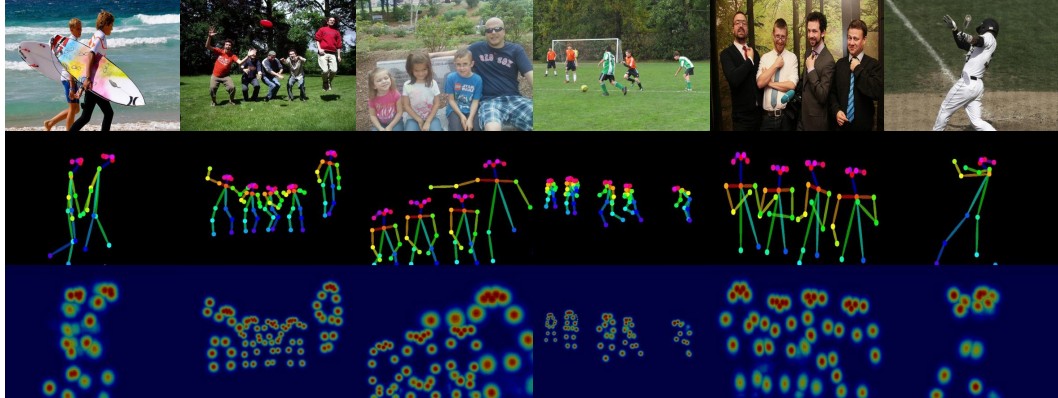

Figure 6: Qualitative visualization of our FlatPose-L model on the COCO test-dev set. The model demonstrates strong localization capabilities across a diverse range of human poses and complex scenes without relying on any upsampling layers.

## C APPENDIX: LIMITATIONS AND FUTURE WORK

While FlatPose demonstrates a highly competitive balance between accuracy and efficiency, its core design choice—the complete abandonment of upsampling layers—introduces specific trade-offs. These are most evident in its slightly lower performance on certain metrics, namely the Average Recall (AR) on the COCO validation set and the Average Precision for hard instances ($AP^{(H)}$) on the CrowdPose benchmark. We attribute these limitations to the inherent challenges of processing spatially coarse feature maps.

**Reduced Recall on COCO val.**  As shown in our combined results table, our FlatPose-L model, despite achieving a higher overall AP, reports a slightly lower AR. Average Recall primarily measures the model's ability to detect all ground-truth keypoints, regardless of precision. Our hypothesis is that the initial aggressive downsampling in the backbone network, which produces the low-resolution feature map for FlatPose, can lead to information loss for certain keypoints. Specifically, keypoints that are very small in scale, have low contrast, or are subtly defined might have their feature signatures "averaged out" or merged with the background in the coarse feature grid. Once this information is lost, even a powerful feature enhancement mechanism like our Transformer head cannot recover it, leading to a failure to detect (a "miss") for these challenging keypoints. In contrast, methods that utilize upsampling maintain a finer spatial grid, which offers a better chance of preserving these weak signals, thus benefiting recall.

**Difficulty in Densely Crowded Scenes.**  On the CrowdPose dataset, which is specifically designed to test robustness in crowded environments, FlatPose-L achieves a competitive overall AP but shows a certain gap in the "Hard" metric ($AP^{(H)}$). This metric evaluates performance on instances with severe occlusion or very close proximity. The primary challenge here is **feature entanglement**. In a low-resolution feature map, keypoints from multiple, closely interacting individuals can be projected onto the same or adjacent feature cells. This creates highly ambiguous feature representations where signals from different people are conflated. While our ICA mechanism is designed to model geometric relationships, it may struggle to first disentangle these aliased features that have lost their precise spatial separation. Upsampling-based methods inherently mitigate this issue by creating more spatial "bins," allowing features from different individuals to remain more spatially distinct and thus easier to process, which is particularly crucial for the most challenging crowded cases.

**Future Work.**  These limitations highlight a clear trade-off between computational efficiency and the ability to resolve spatial ambiguity. Future work could explore hybrid approaches that introduce minimal, highly targeted upsampling or employ more sophisticated feature disentanglement techniques that can operate effectively on coarse feature maps. Developing attention mechanisms specifically designed to deconvolve mixed signals within a single feature token could also be a promising direction to enhance performance in crowded scenes without sacrificing the efficiency of the upsampling-free paradigm.

## D APPENDIX: HYPERPARAMETER DETAILS AND PSEUDOCODE FOR ICA

In response to the reviewer's feedback, this section provides a detailed breakdown of the Implicit Coordinate Attention (ICA) mechanism, first detailing its hyperparameters and then presenting a formal pseudocode description of its forward pass.

### D.1 ICA HYPERPARAMETER CONFIGURATION

Table 6 presents a comprehensive list of hyperparameters for the ICA module, clarifying its architecture and parameterization.

### D.2 ICA PSEUDOCODE

Algorithm 1 provides the step-by-step procedure for the ICA forward pass, detailing how the content-based attention scores are augmented with the dynamically generated semantic bias.

Table 6: Detailed hyperparameter configurations for the Implicit Coordinate Attention (ICA) module.

| Hyperparameter | Symbol | Value / Configuration | Description |
|---|---|---|---|
| Embedding Dimension | $D$ | 256 | The input/output dimension for each token. |
| Number of Heads | $N_h$ | 8 | The number of parallel attention heads. |
| Dimension per Head | $d_k$ | 32 | The dimension for Q, K, V in each head $(D/N_h)$. |
| Semantic Coord. Dim. | $D'$ | 64 | Intermediate dimension for semantic coordinates $(D/4)$. |
| Coord. Gen. Scale | $\gamma$ | $(D')^{-0.5}$ | Fixed scaling factor of $0.125$ for stability. |
| Bias Grid Size | $S$ | 16 | The spatial dimension of the learnable bias grid $(S \times S)$. |
| Bias Grid Feature Dim. | $D_{\text{bias}}$ | 64 | The channel dimension for each point in the bias grid. |
| Interpolator MLP | $\text{MLP}_{\text{interp}}$ | Input: $D_{\text{bias}} + 2 = 66$ Hidden: Linear(128), GELU Output: Linear(8) | The MLP refining interpolated features to produce the final bias for each head. |

---

**Algorithm 1** Implicit Coordinate Attention (ICA) Forward Pass

---

1: **Input:** Token sequence $X \in \mathbb{R}^{L \times D}$, feature map dimensions $H_f, W_f$.
2: **Parameters:** Projection matrices $W_q, W_k, W_v, W_o$; semantic coord. matrices $W_{q,x}, W_{k,x}, W_{q,y}, W_{k,y}$; learnable bias grid $G_{\text{bias}}$; interpolator MLP $\text{MLP}_{\text{interp}}$.

3: # — 1. Standard Content-based Attention Path —
4: $Q_{\text{attn}}, K_{\text{attn}}, V_{\text{attn}} \leftarrow XW_q, XW_k, XW_v$          ▷ Project to Q, K, V for content
5: Reshape $Q_{\text{attn}}, K_{\text{attn}}, V_{\text{attn}}$ to split into $N_h$ heads.
6: $Score_{\text{content}} \leftarrow (Q_{\text{attn}} K_{\text{attn}}^T)/\sqrt{d_k}$          ▷ Shape: $(N_h, L, L)$

7: # — 2. Dynamic Semantic Bias Path —
8: # 2a. Generate semantic coordinates
9: $q_{r,x}, k_{r,x} \leftarrow XW_{q,x}, XW_{k,x}$
10: $c_x \leftarrow \tanh(\gamma \cdot \text{sum}(q_{r,x} \odot k_{r,x}, \dim = -1))$      ▷ Shape: $(L, )$
11: $q_{r,y}, k_{r,y} \leftarrow XW_{q,y}, XW_{k,y}$
12: $c_y \leftarrow \tanh(\gamma \cdot \text{sum}(q_{r,y} \odot k_{r,y}, \dim = -1))$      ▷ Shape: $(L, )$

13: # 2b. Compute relative bias from coordinates
14: $\Delta c_x \leftarrow |c_x[:, \text{None}] - c_x[\text{None}, :]|$      ▷ Relative distance matrix, shape: $(L, L)$
15: $\Delta c_y \leftarrow |c_y[:, \text{None}] - c_y[\text{None}, :]|$
16: $p_{\text{query}} \leftarrow \text{stack}([\Delta c_x - 1, \Delta c_y - 1], \dim = -1)$      ▷ Normalize to $[-1, 1]^2$
17: $f_{\text{interp}} \leftarrow \text{GridSample}(G_{\text{bias}}, p_{\text{query}})$      ▷ Bilinear interpolation
18: $B_{\text{input}} \leftarrow \text{concat}(f_{\text{interp}}, p_{\text{query}})$
19: $B \leftarrow \text{MLP}_{\text{interp}}(B_{\text{input}})$      ▷ Shape: $(L, L, N_h)$
20: $B \leftarrow \text{permute}(B, (2, 0, 1))$      ▷ Final bias, shape: $(N_h, L, L)$

21: # — 3. Combine, Attend, and Output —
22: $Score_{\text{final}} \leftarrow Score_{\text{content}} + B$
23: $A \leftarrow \text{Softmax}(Score_{\text{final}})$      ▷ Attention weights
24: $Y_{\text{heads}} \leftarrow AV_{\text{attn}}$
25: $Y_{\text{merged}} \leftarrow \text{merge\_heads}(Y_{\text{heads}})$
26: $Y \leftarrow Y_{\text{merged}} W_o$
27: **Return:** Output token sequence $Y \in \mathbb{R}^{L \times D}$

---

