# OpenReview forum: "FlatPose: An Upsampling-Free Transformer for Human Pose Estimation"
_ICLR.cc/2026/Conference — Submitted to ICLR 2026_

### Official Review · Reviewer_PNJU · 2025-10-16

**Soundness:** 2
**Presentation:** 3
**Contribution:** 3
**Rating:** 2
**Confidence:** 5

**Summary:**

FlatPose is a novel human pose estimation framework that achieves high efficiency by eliminating the final upsampling layers. The preliminary analysis shows the superior robustness of coordinate classification over heatmap methods at low resolutions. To leverage this, the proposed two-stage architecture first employs a Global Encoding module with an ICA mechanism for  feature modeling, followed by a Targeted Refinement module that uses Salience-Guided Selection to focus computation on critical regions. The method is validated on three major benchmarks, demonstrating a favorable trade-off between high accuracy and practical efficiency.

**Strengths:**

This paper achieve better performance than previous counterparts.

**Weaknesses:**

- Major
1.	The efficiency comparison is based solely on FLOPs, which are insufficient to reflect practical performance. For a meaningful evaluation, it is essential that the authors also report the inference time when models are run on the same GPU

2.	The paper attributes the high performance of FlatPose to its novel design. However, the results in Table 5 suggest that this performance is heavily dependent on the use of a powerful backbone, ConvNeXt-V2. When a more standard backbone, ResNet-50, is used, FlatPose achieves an AP of 73.9 on COCO Val. This result is lower than several competing methods (e.g., SimCC: 75.9, SHaRPose-Small: 74.2, TokenPose-B: 74.7). This raises a concern that the claimed advantages of FlatPose may not be intrinsic to its architectural design but are instead a reflection of the backbone’s strength. To solidly demonstrate the contribution of the FlatPose architecture itself, a more equitable comparison against counterparts using a similar-grade backbone is necessary.

3.	The paper currently lacks comparisons with several relevant end-to-end pose estimation methods that also eliminate up-sampling layers, such as PETR, ED-Pose, and GroupPose. These works represent a significant and relevant branch of literature in simplifying the pose estimation pipeline. Moreover, a key advantage of these end-to-end approaches is their ability to handle multi-person pose estimation in a unified framework. Benchmarking against these methods would provide a more complete and convincing evaluation of FlatPose’s performance and positioning within the field.

4.	The paper claims that FlatPose’s primary contribution is mitigating accuracy degradation under low resolution by eliminating upsampling layers. However, Section 3 (“Methodology”) does not provide a clear theoretical or mechanistic explanation for how this is achieved. The section primarily details incremental design choices based on a Transformer architecture. If the low-resolution performance is primarily attributable to the adoption of the SimCC head—an existing technique—then the novel contribution of the FlatPose architecture itself becomes unclear

- Minor
1.	The citation format is not correct.
2.	There are two many same sentences between L042-L049.

**Questions:**

1.	When evaluate on COCO, which bounding box file do you use? There are two different bbox file: gt bbox and AP56 bbox. Due to FlatPose is a top-down method, incorrect usage can result in a 2-4 point difference in AP. So it is better to directly indicate this in implementation details.

---

### Official Review · Reviewer_f1NW · 2025-10-27

**Soundness:** 2
**Presentation:** 3
**Contribution:** 3
**Rating:** 4
**Confidence:** 4

**Summary:**

This paper aims to address the trade-off between accuracy and efficiency in human pose estimation (HPE), which arises from the reliance of existing methods on computationally expensive upsampling layers. The paper first experimentally demonstrates that the coordinate classification paradigm exhibits stronger robustness to low-resolution feature maps compared to the heatmap paradigm. Based on this insight, the authors propose FlatPose, a completely upsampling-free Transformer framework that operates directly on the low-resolution feature maps from the backbone network. Its core is a two-stage feature enhancement strategy: (1) Global encoding using a novel Implicit Coordinate Attention (ICA) mechanism to learn a dynamic "semantic coordinate system" for modeling geometric relationships; (2) Iterative refinement using salience-guided selection and targeted cross-attention. The authors claim that experiments on benchmarks like COCO show FlatPose achieves a state-of-the-art (SOTA) balance between accuracy and efficiency.

**Strengths:**

1. Clear and Well-Argued Motivation: The paper's starting point is clear, using experiments (Fig 1b) to convincingly show the robustness of coordinate classification methods at low resolutions, providing solid empirical evidence for the "upsampling-free" direction.
2. Novel Architectural Design (ICA Mechanism): Building on the validated feasibility of upsampling-free approaches, the proposed ICA mechanism is a novel attempt to model geometric relationships within the Transformer framework by dynamically generating attention bias based on learned relative semantic distances.
3. Significant Computational Efficiency: The model itself demonstrates excellent efficiency. For instance, FlatPose-L achieves competitive accuracy (77.4 AP) on COCO val with significantly reduced GFLOPs (4.9 GFLOPs), showing potential for resource-constrained scenarios.
4. Thorough Internal Ablation Studies: Table 5 provides a systematic analysis validating key design components (e.g., the necessity of ICA, the importance of coordinate classification, redundancy of upsampling layers), offering strong internal support for the design choices.

**Weaknesses:**

1. (Major Concern) SOTA Comparisons Lack Fairness due to Confounding Variables: As shown in Table 2, FlatPose-L uses ConvNeXtV2-T, while the compared RTMPose-l uses CSPNeXt-l. These backbone architectures have inherent performance differences. Therefore, it's impossible to determine how much of FlatPose-L's performance gain comes from its novel Transformer head versus the potentially stronger backbone. This comparison fails to effectively isolate variables and cannot prove the superiority of the FlatPose head itself.
2. Misleading Result Presentation Compromises Scientific Rigor: In Table 3, FlatPose-L's parameter count of 37.4M is bolded, which is clearly smaller than ViTPose-B's 89.9M but larger than TokenPose-L/D24's 29.8M. The table caption explicitly states, "Best results for each metric are in bold." The authors have clearly violated their own stated convention. This practice can easily mislead readers into believing FlatPose is optimal across all bolded metrics, severely damaging the paper's scientific rigor and credibility.
3. AR limitation: These performance deficits, attributed by the authors to information loss for challenging keypoints (small, low-contrast, occluded/entangled) on coarse feature maps, represent a fundamental trade-off of the upsampling-free paradigm. By placing this analysis outside the main text, the paper fails to adequately contextualize its efficiency gains against the associated compromises in detection capability for difficult cases, potentially obscuring a complete understanding of the method's performance profile.

**Questions:**

1. Could you provide fair comparison results for your FlatPose head against ViTPose head, RTMPose head (or SimCC head) under the same backbone ? Alternatively, please detail the specific pre-training dataset and method used for your ConvNeXtV2 backbone and justify its comparability to the methods being compared against ?
2. In Table 3, parameter count values that are not the column minimums are bolded, contradicting your caption "Best results for each metric are in bold." To ensure scientific rigor and clear communication, would you be willing to correct this issue in the final version by strictly adhering to the caption and only bolding the absolute best value in each column?
3. The ICA module computes a dense L*L bias matrix. While L is relatively small in this work , could you briefly analyze the computational and memory overhead of this step? How does its scalability fare as the feature map resolution increases? Under what circumstances might it become a bottleneck?

---

### Official Review · Reviewer_ud2Y · 2025-10-28

**Soundness:** 2
**Presentation:** 2
**Contribution:** 2
**Rating:** 2
**Confidence:** 4

**Summary:**

This paper proposes FlatPose, a Human Pose Estimation approach that removes the final upsampling layer. The authors first conduct a preliminary analysis to show that coordinate classification-based methods are more robust than heatmap-based methods when the feature map resolution is small. Then, they construct a two-stage hierarchical framework (FlatPose) to enhance the vision features. In the Global Encoding stage, they introduce the ICA mechanism to modulate the attention map, and in the Targeted Refinement stage, they introduce the Salience Guided selection strategy to refine the critical features.

**Strengths:**

1. FlatPose achieves a good balance between accuracy and theoretical efficiency on the performance of COCO, MPII, and CrowdPose, when compared with many SOTA methods.
2. This study on the model design with low-resolution representation can be meaningful for exploring efficient and powerful human pose estimators.

**Weaknesses:**

1.  Poor Writing. In Lines 043-048, the statements are completely repetitive.
In Line 213, the authors have not explained what S is in G_bias.
Too many duplicated and meaningless equations. For example, in e.q.(6) and e.q.(7), the only difference between p_query and delta_C is p_query = delta_C - 1. The two equations can easily be assembled. What’s more, e.q.(1) is only a 3x3 conv layer, e.q.(8) is just a bilinear interpolation operator, e.q.(11) is a pytorch gather operator. E.q.(12) is merely a Cross Attention layer; there is no need to waste valuable space with equations.

2. The authors designed the preliminary experiments to show that coordinate classification-based methods are more robust than heatmap-based methods when the feature map resolution is small. But the same conclusion has been evaluated in SimCC (ECCV 2022), without showing more insight.

3. Unfair Comparison. FlatPose uses a strong backbone ConvNeXt-V2. According the Table 5, part A, when using ResNet-50 as backbone, the AP on COCO-Val is 73.9, which demonstrates that the accuracy of FlatPose depends on the powerful backbone. When removing the ConvNeXt-V2, the performance on COCO-val is lower than many counterparts, such as SimCC (75.9), RTMPose (74.8) ViTPose-B (75.7) and TokenPose-L/D24 (75.8)

**Questions:**

What is the core spirit of the concept of “semantic coordinate system” as described in the paper? Why does learning such a capability matter?

---

### Official Review · Reviewer_x4LG · 2025-10-31

**Soundness:** 3
**Presentation:** 4
**Contribution:** 3
**Rating:** 6
**Confidence:** 4

**Summary:**

This paper proposes FlatPose, a new transformer architecture for human pose estimation that aims to improve efficiency by performing all computations on low-resolution feature maps. The authors argue that this upsampling-free design leverages the inherent robustness of coordinate classification-based methods to feature resolution changes, a claim supported by their preliminary experiments. To handle the challenge of decoding pose from coarse features, the paper introduces two core components: an Implicit Coordinate Attention (ICA) mechanism to model geometric relationships, and a Salience-Guided Selection module to focus computation on critical feature regions.

**Strengths:**

The paper is well written, with a clear structure, informative figures, and comprehensive tables. The proposed FlatPose model is efficient and demonstrates strong performance, supporting the authors’ claim that feature upsampling is unnecessary for human pose estimation.

**Weaknesses:**

The authors argue that coordinate classification-based methods are more robust to feature resolution. Preliminary experiments in Figure (b) show a significant performance drop for heatmap-based methods when the feature resolution is reduced. However, according to Table 5, part A(3), even at the lowest resolution and with heatmap prediction, FlatPose-B still achieves relatively good results—the performance degradation is much more stable than what is shown in Figure (b). This inconsistency makes the paper’s main contribution somewhat unclear.

**Questions:**

The authors may need to clarify the specific roles of different components in their method. Both the Implicit Coordinate Attention mechanism and the Salience-Guided Selection module appear unnecessary based on the current presentation.
I’m curious about what the attention bias looks like, which “concrete” problem it solve? its working mechanism remains somewhat abstract.

---

### Meta-Review · Area_Chair_CJn7 · 2026-01-12

**Summary:**

Reviewers agree that FlatPose addresses an important and timely problem in human pose estimation by exploring an upsampling-free design aimed at improving efficiency. The motivation—supported by experiments showing the robustness of coordinate classification at low resolution—is generally well received, and the proposed architecture demonstrates a strong accuracy–efficiency trade-off, particularly in terms of GFLOPs. The ICA mechanism and salience-guided refinement are viewed as interesting design choices, and internal ablations provide evidence that these components contribute to the reported performance. Overall, the paper shows potential for efficient pose estimation in resource-constrained settings.

However, reviewers consistently raise serious concerns about experimental fairness and clarity of contribution. Reported gains are confounded using a strong ConvNeXt-V2 backbone, and performance drops substantially with more standard backbones, making it difficult to attribute improvements to the FlatPose architecture itself. Evaluation practices are viewed as incomplete or misleading (e.g., selective metrics, lack of same-GPU latency, table formatting issues), and comparisons to relevant upsampling-free or end-to-end methods are missing. Additionally, reviewers note that the paper lacks a clear theoretical or mechanistic explanation for its core claims and suffers from writing and presentation issues in parts.

There are four reviewers, with one positive score and three negative scores. No rebuttals are provided.
Thus, questions of three negative reviewers are not solved.

**Reviewer Scores:**

no

---

### Decision · Program_Chairs · 2026-01-26

Reject